# Serpinin in the Skin

**DOI:** 10.3390/biomedicines10010183

**Published:** 2022-01-16

**Authors:** Cristina Fraquelli, Jasmine Hauzinger, Christian Humpel, Maria Nolano, Vincenzo Provitera, Vinay Kumar Sharma, Peng Loh, Zenon Pidsudko, Georgios Blatsios, Josef Troger

**Affiliations:** 1Department of Ophthalmology, Medical University of Innsbruck, Anichstraße 35, 6020 Innsbruck, Austria; cristina.fraquelli@tirol-kliniken.at (C.F.); jasmine.hauzinger@tirol-kliniken.at (J.H.); georgios.blatsios@i-med.ac.at (G.B.); 2Laboratory of Psychiatry and Experimental Alzheimers Research, Department of Psychiatry, Medical University of Innsbruck, Anichstraße 35, 6020 Innsbruck, Austria; christian.humpel@i-med.ac.at; 3Skin Biopsy Laboratory, Department of Neurology, Instituti Clinici Scientifici Maugeri IRCCS, Via Maugeri 4, 27100 Pavia, Italy; maria.nolano@icsmaugeri.it (M.N.); vincenzo.provitera@icsmaugeri.it (V.P.); 4Department of Neurosciences, Reproductive Sciences and Odontostomatology, University Federico II of Naples, 80100 Naples, Italy; 5Section of Cellular Neurobiology, Eunice Kennedy Shriver National Institute of Child Health and Human Development, National Institutes of Health, Bethesda, MD 20892, USA; vinay.sharma@nih.gov (V.K.S.); lohp@mail.nih.gov (P.L.); 6Department of Animal Anatomy, Faculty of Veterinary Medicine, University of Warmia and Mazury in Olsztyn, Oczapowskiego 13, 10-719 Olsztyn, Poland; zenekp@uwm.edu.pl

**Keywords:** serpinin, skin, dorsal root ganglia, substance P

## Abstract

The serpinins are relatively novel peptides generated by proteolytic processing of chromogranin A and they are comprised of free serpinin, serpinin-RRG and pGlu-serpinin. In this study, the presence and source of these peptides were studied in the skin. By Western blot analysis, a 40 kDa and a 50 kDa protein containing the sequence of serpinin were detected in the trigeminal ganglion and dorsal root ganglia in rats but none in the skin. RP-HPLC followed by EIA revealed that the three serpinins are present in similar, moderate amounts in rat dorsal root ganglia, whereas in the rat skin, free serpinin represents the predominant molecular form. There were abundant serpinin-positive cells in rat dorsal root ganglia and colocalization with substance P was evident. However, much more widespread distribution of the serpinins was found in dorsal root ganglia when compared with substance P. In the skin, serpinin immunoreactivity was found in sensory nerves and showed colocalization with substance P; as well, some was present in autonomic nerves. Thus, although not exclusively, there is evidence that serpinin is a constituent of the sensory innervation of the skin. The serpinins are biologically highly active and might therefore be of functional significance in the skin.

## 1. Introduction

The serpinins are relatively novel neuropeptides that are generated by proteolytic processing of chromogranin (Cg) A [1,2]. CgA belongs to the family of granins, which are the acidic proteins of secretory granules in a variety of endocrine and neuroendocrine cells as well as cells of the central and peripheral nervous system, where they are stored in large dense core vesicles. They additionally comprise CgB and secretogranin (Sg) II [3,4]. There are several consecutive pairs of basic residues present in the primary amino acid sequence of the granins and these sites are targets of prohormone convertases (PCs). There exist three main PCs, namely PC1, PC2 and furin, and these PCs are enzymes which cleave the granins in a tissue specific manner [5]. Several fragments are generated by proteolytic processing of the granins, such as catestatin, pancreastatin, vasostatin, serpinin or GE-25 by proteolytic processing of CgA, secretolytin or PE-11 by proteolytic processing of CgB and secretoneurin by proteolytic processing of SgII (review, see [6]). A conceptional schematic drawing of the cleavage sites of the chromogranins and the resultant peptides has been shown by the authors previously (see [6] again).

There are just a few studies about serpinin found in the literature. In particular, serpinin secreted from corticotroph cells has initially been shown to up-regulate the expression of protease nexin-1 (PN-1), a protease inhibitor which then acts to increase CgA levels in the Golgi apparatus, thereby promoting granule biogenesis [1,7]. Thereafter, two further forms have been identified, in particular pyroglutaminated (pGlu)-serpinin and the C-terminal extended form serpinin-RRG [1,8]. Especially pGlu-serpinin is now well known to have neuroprotective activity since it prevented cell death induced by oxidative stress in pituitary AtT-20 cells and cultured rat cerebral cortical neurons [2]. Serpinin-RRG is present in higher amounts than pGlu-serpinin in the rat heart [9] and, furthermore, both serpinin and pGlu-serpinin exert positive inotropic and lusitropic effects via a ß1-adrenergic receptor/adenylate cyclase/cyclic adenosine monophosphate (cAMP)/protein kinase A (PKA) pathway [9]. Previously, pGlu-serpinin has been shown to protect the normotensive and hypertensive heart from ischemic injury during reperfusion by reducing myocardial infarct size [10] as well as depressing myocardial performance in lower vertebrates such as fish and frogs [11].

Serpinin has recently been found to be a constituent of the sensory innervation of the eye [12] and dental pulp [13]. In the present study we aimed to investigate the distribution of the serpinins in another tissue, particularly the skin, including the demonstration of its presence in sensory nerves there as well.

## 2. Materials and Methods

### 2.1. Animals

Male Sprague–Dawley and male Wistar rats weighing approximately 200 to 250 g each were used for Western blots, RP-HPLC and immunofluorescence studies in dorsal root ganglia (DRG). The animals were purchased from Charles River (Sulzfeld, Germany) and the Sprague-Dawley rats were housed in stainless steel cages. They were fed standard rat chow and tap water ad libitum and were held in a room on a 12-h light-dark cycle with an ambient temperature of 22 °C ± 1 °C and treated according to the standards of Innsbruck’s Medical University. The Wistar rats were housed and treated in accordance with the rules approved by the local Ethics Commission in Olsztyn (affiliated to the National Ethics Commission for Animal Experimentation, Polish Ministry of Science and Higher Education).

### 2.2. Western Blot

Western blots were performed as described by us previously [14]. Briefly, the tissue was dissolved in 200 µL ice-cold PBS with a protease inhibitor cocktail (P8340, Sigma, Hamburg, Germany), homogenized by using an ultrasonic device (Branson sonifier 250, Danburry) and centrifuged at 16,000× *g* at 4 °C for 10 min. Protein was determined by the Bradford assay. The non-denatured supernatants (25 µg) were loaded onto a 10% Bis-Tris polyacrylamide gel (Invitrogen, Waltham, MA, USA) and were electrophoresed for 25 min at 200 V. Samples were electrotransferred to nylon PVDF (Immobilon-PSQ membranes (Millipore, Burlington, MA, USA) for 90 min at 30 V with 20% methanol blotting buffer (Invitrogen). Blots were blocked for 30 min with blocking buffer, then incubated overnight at 4 °C with the primary serpinin antibody (1:1000; gift from Peng Loh, National Institutes of Health, Bethesda, MD, USA; polyclonal antibody, raised in rabbits) or overnight at 4 °C with a rabbit anti-actin antibody (1:1000, Sigma-Aldrich, St. Louis, MO, USA). Blots were washed and incubated with alkaline phosphatase-conjugated anti-rabbit antibodies for 30 min at room temperature. After washing, blots were incubated (free-floating) in CDP-Star chemiluminescent substrate solution (Invitrogen) for 10 min and the signal was visualized (exposure time 1200 s) with a cooled CCD camera (SearchLight, Thermoscience).

### 2.3. RP-HPLC with EIA

Extraction from pooled thoracic DRGs and skin was performed by sonication in distilled water for 10 s and then by boiling for 10 min. After centrifugation (20 min at 12,000× *g*), the supernatant was dried by vacuum centrifugation. The heat extracted tissues were dissolved in 100 µL of 0.25% acetic acid and were processed for 2 min using silica beads. The samples were then boiled at 95 °C for 10 min and centrifuged for 15 min at 17,000× *g*. Supernatants were saved in a fresh tube and the pellet was re-extracted with 100 µL of 0.25% acetic acid and combined with the first extraction. The supernatants were then passed through the 10 kDa filter device (Micron YM-10, Millipore, Billericia, MA, USA) and lyophilized. The dried eluates were reconstituted with 200 µL 0.1% TFA and separated by HPLC on a 4.6 × 250 mm 5 µm reverse phase Jupiter C18 column (Phenomenex, Torrance, CA, USA). Buffer A was 0.1% TFA and buffer B was 80% acetonitrile/0.1% TFA and the gradient was 20% to 100% B in 35 min. One ml fractions were collected and lyophilized. Thereafter the serpinin peptides were detected in the HPLC purified samples by a direct enzyme-linked immunoassay (EIA).

### 2.4. Double Immunofluorescence for Serpinin/Substance P (SP) in Rat DRG

The study was carried out on four sexually mature rats of the Wistar breed weighing approximately 250 g each. The animals were deeply anaesthetized with thiopental sodium intraperitoneal injection (30 mg/kg of body weight), sacrificed and transcardially perfused with 0.2–0.3 L of 4% ice-cold buffered paraformaldehyde (pH 7.4). Subsequently, bilateral DRG from cervical (C), thoracic (TH) and lumbar (L) spinal regions were collected and postfixed by immersion in the same fixative for 30 min, rinsed with phosphate buffer (pH 7.4) and transferred to and stored in 30% buffered sucrose solution (pH 7.4) until further processing. Transversal 8-µm-thick cryostat serial sections of each collected ganglion were cut and mounted on chrome alum-gelatine-coated slides, air dried and finally processed for the double immunofluorescence method. The sections were washed 3 × 10 min in PBS and blocked (60 min, room temperature (RT)) with a blocking solution containing 10% of normal goat serum (NGS, Cappel, Warsaw, Poland) and 0.25% of Triton X-100 (Sigma, USA) dissolved in PBS and then incubated overnight in RT with antibodies diluted in blocking solution (serpinin antibody, gift from Peng Loh, dilution: 1:500) (SP antibody, mouse, Acris, Cat. No. MO15094, USA, dilution: 1:5000). After incubation with primary antibodies, the sections were washed 3 × 10 min in PBS and were further incubated with corresponding secondary antibodies for 1 h in RT (Alex Fluor 555 goat anti-mouse IgG1, Invitrogen Molecular Probes, Eugene, Oregon, USA, Cat. No. A21127, dilution: 1:100) (Alexa Fluor 488 goat anti-rabbit IgG, Invitrogen Molecular probes, Eugene, Oregon, USA, Cat no. A11008, dilution. 1:1000). After the incubation, the sections were washed 3 × 10 min in PBS, cover slipped with buffered glycerol and studied and photographed under a confocal microscope (Zeiss LSM 710).

### 2.5. Double Immunofluorescence for Serpinin/Protein Gene Product (PGP) 9.5 and Serpinin/SP in the Human Skin

Three mm punch biopsies from four healthy subjects (two male, two female; age 58.7 ± 17.2 years) were obtained from hairy skin (thigh and distal leg) and glabrous skin (fingertip) under local anesthesia after signing the informed consent forms of IRB authorization (“Istituto di Ricovero e Cura a Carattere Scientifico (IRCCS) Pascale” Ethical Commmittee ref. 04/2018). Skin samples were cut at 50 µm using a sliding freezing microtome (Leica 2000R) and sections were processed for indirect immunofluorescence using previously described techniques [15]. Skin sections were incubated with rabbit polyclonal anti-serpinin antibody (a gift from Peng Loh, dilution 1:5000) and double marked with either mouse anti PGP 9.5 (Amsbio, Abingdon, UK; dilution 1:400) as pan neuronal marker or mouse anti SP (Neuromics antibodies, Edina, MN US; dilution 1:200) as a marker for sensory fibers. Skin sections were then incubated with species specific secondary antibodies coupled with Cy3 and Cy2 fluorophores (Jackson Immunoresearch at dilution 1:200). Digital images were acquired using a non-laser confocal system (Apoptome Zeiss, Jena, Germany) equipped with 20× and 40× Plan Apochromat objectives.

## 3. Results

### 3.1. Western Blot Analysis of Immunoreactive Forms of Serpinin in Trigeminal Ganglia (TG), DRG and Skin

Western blots were performed to evaluate the molecular forms of serpinin in various tissues using a highly sensitive serpinin antibody. The tissues analyzed were the TG, DRG and skin, and the characterization of the immunoreactivities was performed in duplicate. The result is illustrated in Figure 1. In the TG, there were two bands present, in particular one at approximately 40 kDa and the other one at approximately 50 kDa representing larger molecular forms containing the sequence of serpinin. In the DRG, the results were completely identical to the TG, in particular two bands at approximately 40 kDa and 50 kDa. By contrast in the skin there was no unequivocal specific band present.

### 3.2. Analysis of Serpinin Peptides in Rat Thoracic DRG and Skin

The molecular forms of serpinin peptides in rat thoracic DRGs and skin were characterized by RP-HPLC followed by EIA and the results are illustrated in Figure 2. With respect to DRGs, there were two peaks present, in fraction 29 and fraction 33, representing serpinin-RRG and free serpinin, respectively, and a shoulder in fraction 34 representing pGlu-serpinin. In the skin, the immunoreactivities in fraction 29 and 34 were about equal, similar to DRGs, but the peak in fraction 33 representing free serpinin was much higher than in DRGs.

### 3.3. Immunofluorescence Studies of Serpinin and SP in Rat Thoracic DRG

The presence and distribution of serpinin-like immunoreactivity (LI) was studied by immunofluorescence histochemistry in rat thoracic DRG and, furthermore, colocalization studies with SP-IR were performed in this tissue. The results are illustrated in Figure 3. Numerous serpinin-like immunoreactive cells were stained in these ganglia, particularly shown as green fluorescence within the cytoplasm of the cells as seen in images A, B, C, D and F of the figure. The cells were small- and medium-sized featuring a diameter of 20–30 µm and were arranged mainly as clusters whereas non-neuronal cells or tissues were not stained. Double-immunofluorescence studies revealed colocalization of serpinin-LI (Figure 3D) with SP-IR (Figure 3E) (indicated by arrows in Figure 3D,E, arrows in superimposed image in Figure 3F), but most serpinin-like immunoreactive cells were devoid of SP-IR, indicating a more widespread expression of serpinin-LI when compared with SP in thoracic DRG.

### 3.4. Immunofluorescence Studies of Serpinin in Human Skin

The results of the immunofluorescence studies in human skin are illustrated in Figure 4. Serpinin immunoreactivity (IR) has been found in the somatic component of cutaneous fibers, featuring colocalization with SP (Figure 4A) and PGP 9.5 (Figure 4B) in the subepidermal neural plexus and in the reticular dermis. Furthermore, serpinin IR has also been observed in autonomic sympathetic fibers innervating arrector pili muscle (Figure 4C) and, more sparsely, in sweat glands (Figure 4D).

## 4. Discussion

The most important finding of this study is that serpinin is a constituent of the sensory innervation of the skin as are the CgB-derived peptides PE-11 and secretolytin [16]. Furthermore, certain granin-derived peptides have been demonstrated to be a constituent of the sensory innervation of the eye, in particular the SgII-derived neuropeptide secretoneurin [17], the CgB-derived peptide PE-11 [18] and the CgA-derived peptides GE-25 [19], catestatin [14] and serpinin [12]. Moreover, secretoneurin and PE-11 [20], as well as serpinin [13], have been shown to be constituents of the sensory innervation of the dental pulp. All of these findings indicate that the CgA-derived peptide serpinin and presumably also other granin-derived peptides are not only constituents of the sensory innervation of the eye and dental pulp, but also of the skin. Interestingly, serpinin IR was observed in autonomic nerves as well, furnishing structures such as arrector pili muscle and sweat glands, which indicates that serpinin is not exclusively a constituent of sensory nerves.

It must be emphasized that there were differences in the band profile in Western blots between TG and DRG, especially for PE-11, indicating that the processing of CgB might be different in sensory ganglia [16]. This seems not to be the case for CgA since the band profile was completely identical in TG and DRG in this study, demonstrating that the processing of CgA leads to generation of identical fragments containing the sequence of serpinin in these ganglia. By contrast, in DRGs there are both free serpinin and serpinin-RRG and less pGlu present, whereas in the rat TG, only free serpinin was detected [12], demonstrating that the processing of CgA to the serpinins is different in sensory ganglia. However, in the skin free serpinin represents the predominant molecular form which is equal to the human dental pulp [13] indicating that mainly this serpinin is present in sensory innervated target tissues.

Previously, it has been implicated that CgA might become proteolytically processed either already in respective sensory ganglia [12,13,14] or by contrast during anterograde axoplasmatic transport [19]; the former observation has been fully confirmed in this study. It can be concluded that there is now unequivocal evidence that CgA becomes proteolytically processed already in sensory ganglia. However, the presence of signals in TG and DRG but the absence of a signal in the skin in Western blots indicates that at least these fragments are yet proteolytically processed during anterograde axonal transport. It must be emphasized that there might be some faint bands in the skin in the Western blot. Because of this reason the authors repeated the Western blot just for the skin which revealed absence of specific bands (not shown). Therefore, it was concluded that there are no specific bands present in the skin. Finally, the serpinin-positive nerve fibers in the skin must be predominantly unmyelinated C-fibers and/or myelinated Aδ-fibers since these fibers originate from small to medium-sized ganglion cells in sensory ganglia [21]. The expression of serpinin-LI in small- to medium-sized ganglion cells has been demonstrated and confirmed now in our study.

There were no functional experiments performed in this study but the serpinins are biologically highly active (see Section 1) and therefore a brief discussion of their potential functional significance in the skin must be included. There are three main functional topics where serpinin might be involved, in particular in granulogenesis, neuroprotection and in ß-adrenergic agonism. Firstly, serpinin might promote granulogenesis in sensory nerves innervating the skin since free serpinin particularly is present in such nerves. With respect to neuroprotection, there are only two further neuropeptides that exert such a strong neuroprotective potency; in particular pituitary adenylate cyclase-activating polypeptide (PACAP) and vasoactive intestinal polypeptide (VIP). Although to our knowledge no direct neuroprotective effects have been reported from these neuropeptides in the skin, they regulate immunity which might be associated with protective and neuroprotective effects [22]. Furthermore, PACAP is involved in the development of inflammatory pathology such as psoriasis and also has anti-allergic effects in a model of contact-dermatitis [23]. Thus, serpinin might be involved in similar physiological and pathological conditions. Finally, the serpinins are ß-adrenoreceptor agonists (see above in the Section 1) and may act on the ß2-adrenoreceptor as well. Beta2-adrenoreceptor is known to be the main mediator of vasodilation as a result of vascular smooth muscle cell relaxation [24,25,26]. Hence serpinin might be involved in physiological ß2-adrenergic-associated vasodilation in the skin as well as in pathophysiological vasodilation, in particular in neurogenic inflammation.

In conclusion, this peptide is a constituent of the sensory innervation of the skin and thus seems to be involved in sensory transmission. Since serpinin is biologically highly active, it might be of functional significance in this tissue, which must be explored in the future.

## Figures and Tables

**Figure 1 biomedicines-10-00183-f001:**
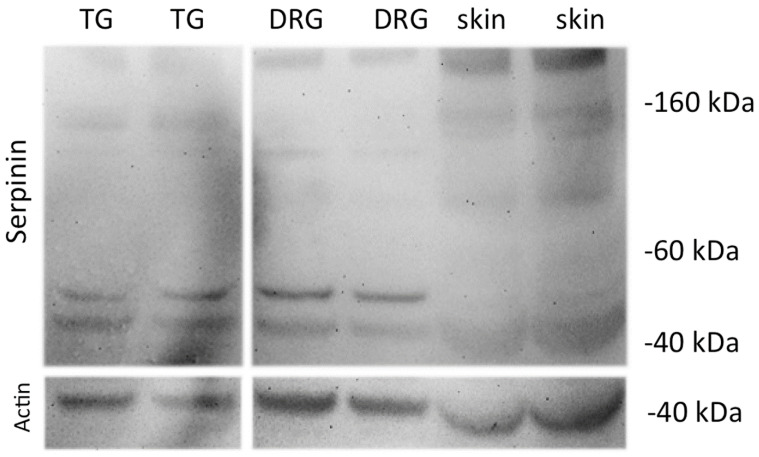
Western blot analysis of serpinin immunoreactivities in rat TG, rat DRG and rat skin. Note the presence of two strong bands at 40 kDa and 50 kDa in both TG and DRG representing two larger molecular forms containing the sequence of serpinin, whereas no bands were observed in the skin. Actin represents an internal standard at 42 kDa as a control.

**Figure 2 biomedicines-10-00183-f002:**
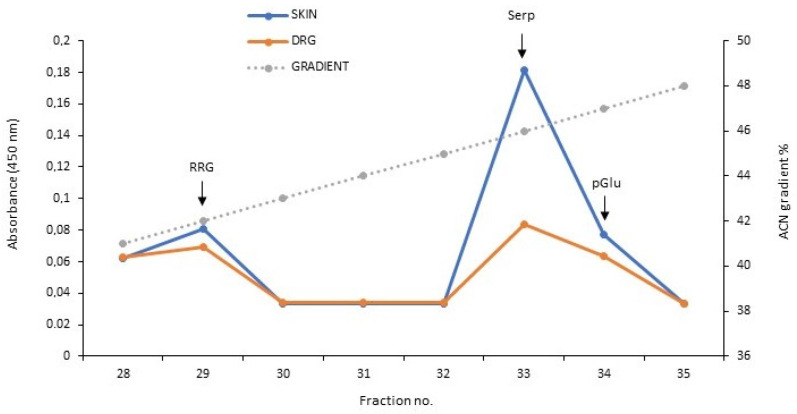
RP-HPLC and EIA for serpinin-like immunoreactivity from extracts of rat DRG and skin. In DRGs (brown line), there were two peaks present, in fraction 29 and 33 and a shoulder in fraction 34 representing serpinin-RRG (“RRG”), free serpinin (“Serp”) and pGlu-serpinin (“pGlu”), respectively. In the skin (blue line), the immunoreactivities in fraction 29 and 34 were approximately equal to that in DRGs, but the immunoreactive peak in fraction 33 was much higher than in DRGs. The elution positions of these three serpinins are indicated by arrows.

**Figure 3 biomedicines-10-00183-f003:**
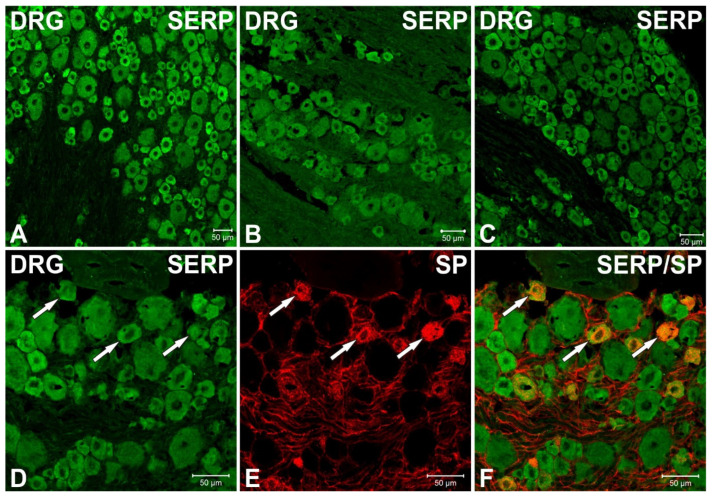
Serpinin-LI in rat thoracic DRG showing double-immunofluorescence of serpinin/SP. Many serpinin immuno-positive cells were observed in these ganglia (green), the cells featured a diameter of 20–30 µm and were arranged mainly in clusters (**A**–**C**). Furthermore, double-immunofluorescent studies of serpinin ((**D**), green) with SP-IR ((**E**), red) revealed colocalization (indicated by arrows in (**D**,**E**), arrows in the superimposed image (**F**), orange) but most serpinin positive cells were devoid of SP-IR. Scale bars represent 50 µm.

**Figure 4 biomedicines-10-00183-f004:**
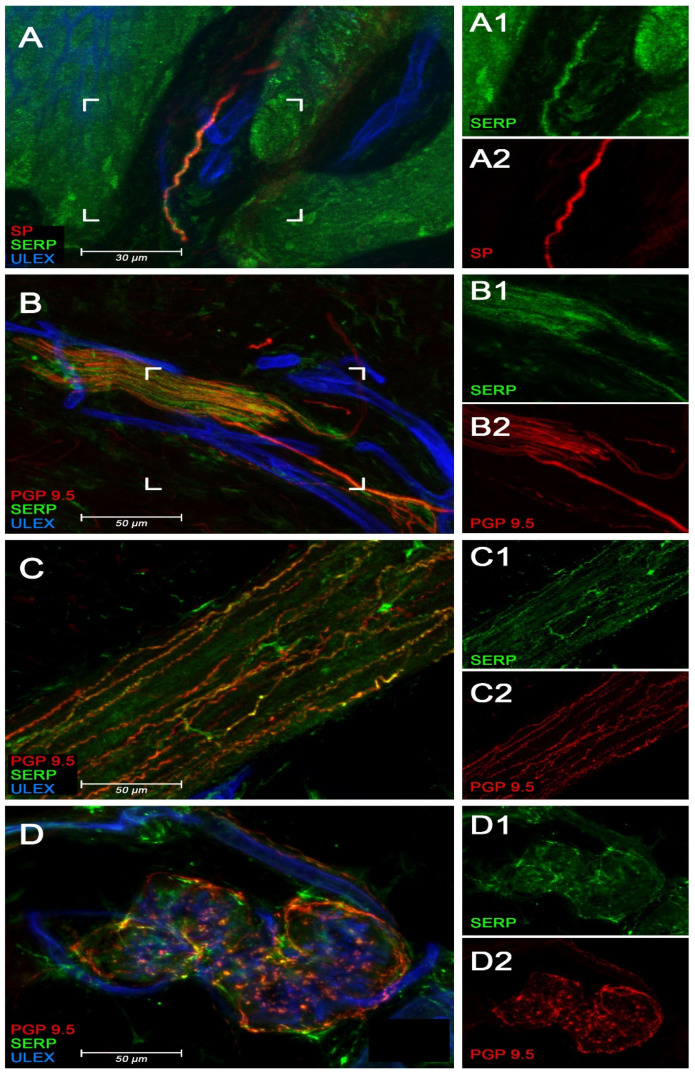
Confocal images demonstrating the presence of serpinin IR within nerve fibers in human skin from control subjects. On the left, the merged images (**A**–**D**) show colocalization of serpinin with neural markers. On the right, the unmerged images (**A1**,**A2**,**B1**,**B2**,**C1**,**C2**,**D1**,**D2**) allow to better appreciate the different stainings. (**A**): serpinin (green) colocalization with SP (red) in a sensory fiber within a dermal papilla. (**B**): serpinin (green) colocalization with the pan neuronal marker PGP 9.5 (red) in the fibers included in a nerve bundle in the reticular dermis. (**C**): serpinin (green) colocalization with PGP 9.5 (red) in pilomotor sympathetic nerves from an arrector pili muscle. (**D**): serpinin (green) colocalization with PGP 9.5 (red) in sudomotor sympathetic nerves around sweat tubules. Scale bar is 30 µm in (**A**) and 50 µm in (**B**–**D**). SP = substance P; PGP 9.5 = Pan neuronal marker protein gene product 9.5; ULEX = Ulex europaeus agglutinin 1.

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
