# Peer review of "Serpinin in the Skin"

_biomedicines, 2022, doi:10.3390/biomedicines10010183_

Round 1

Reviewer 1 Report

The authors report the main (cellular) sites of serpinin types using immunochemistry and fluorescent images. According to the authors, the localization of the serpinins, the bioactive peptide products of proteolytic processing of chromogranin, indicates a biological role of cellular protection in the sensory neurons.

Before acceptation, it is recommendable to improve some points in the manuscript, as follow:

  • It is recommendable to describe all abbreviations the first time they appear in the text, like SP, PGP, AT-20 (?) cells, PKA pathway, pGlu, IRCCS…
  • It will be attractive if the authors include a schematic draw with the sequence/protein domains of granin and all known/possible proteolytic peptide products. Indicating cleavage sites of chromogranin, and the resultant peptides with their sizes and molecular masses. This schematic diagram makes the immunodetection results more understandable and remarkable;
  • If the journal requires it, it is recommendable to uniformize the company names, city, and country.
  • In M&M, Who is Mr./Ms. Peng Loh?
  • It is essential to specify the nature of antibodies (polyclonal x monoclonal) and how the serpin antibody was prepared.
  • What were the criteria to affirm that serpinin antibody was highly sensitive? It is interesting to discuss this affirmation.
  • The authors affirm that "The most important finding of this study is that serpinin is a constituent of the sen-1 sory innervation of the skin similar to the presence of CgB-derived peptides PE-11 and 2 secretolytin...". What is the difference between this study with the "in press" article, reference 14 by the same authors? It is recommendable to authors to detail this point regarding the findings in both studies.
  • Specify what PACAP and VIP peptides are.
  • Are there recently published articles regarding serpinin and granin-derived peptides? The cited references are a bit old, and it is recommendable the authors discuss this point or cite complementary new references.

Reviewer 2 Report

The title of the article should reflect the overall outcome or the goal with approaches. The hypothesis and the objectives of this article are not quite clearly depicted in the introduction. The readers may find a hard time finding the rationale for this article.

Page 8, line 36/37 seemed poor justification.

Figure1 looks like there are faint bands for skin, there should have an explanation if they are not serpinin, should have mentioned the trial number. 

Should have a concise conclusion that focuses on the importance of the outcomes and subsequent future steps. 

Suplementary section should be well organized.
